# Evaluation of Fecundity, Fertilization, Hatching, and Gonadosomatic Index of Exotic *Clarias gariepinus* (Burchell, 1822) and Native *Clarias macromystax* (Gunther, 1864) under Semi-Arid Conditions of Nigeria

**DOI:** 10.3390/ani13111723

**Published:** 2023-05-23

**Authors:** Yuzine B. Esa, Abdulrahman Muhammad Dadile, Fadhil Syukri, Annie Christianus, Mohammad Y. Diyaware

**Affiliations:** 1Department of Aquaculture, Faculty of Agriculture, University Putra Malaysia (UPM), Serdang 43400, Selangor Darul Ehsan, Malaysia; 2Department of Biological Science, Yobe State University, Damaturu 600213, Nigeria; 3Department of Fisheries, Faculty of Agriculture, University of Maiduguri, Maiduguri 600004, Nigeria

**Keywords:** African catfish, fertilization, hatching success, growth rate, heterosis, survival rate

## Abstract

**Simple Summary:**

The study evaluates the hybridizations in two African catfish, i.e., *C. gariepinus* and *C. macromystax*, using artificial reproduction. Reproductive and growth performance were evaluated. The results indicated that both species possess a similar gonadosomatic index. The parent *C. gariepinus* possesses higher male reproductive quality than *C. macromystax*. Fecundity was higher in the female parent of *C. macromystax* and lower in the *C. gariepinus* species. The highest fertilization rates and hatching rates were achieved in the hybrid cross of *C. macromystax* × *C. gariepinus*, which also recorded the lowest deformed larvae. Survival rates at the larval stage were more than 80% with the highest record of survival in the hybrid *C. macromystax* × *C. gariepinus*. However, survival rates in the fry stage were lower than 70%. The hybrid cross *(C. macromystax* × *C. gariepinus*) showed the best growth performance, which was similar to the parental cross (*C. gariepinus* × *C. gariepinus*). The study revealed the potentials of the hybrid *C. macromystax* × *C. gariepinus*, and hybridizations of *C. macromystax* × *C. gariepinus* (*Cm* × *Cg*) are indeed possible and proved to have better growth and survival rate under semi-arid conditions, which would contribute significantly to the improvement of *C. macromystax* production in captivity.

**Abstract:**

The study evaluates the hybridizations between two African catfish, *C. gariepinus* and *C. macromystax,* using artificial reproduction. Fecundity and gonadosomatic index were assessed, and growth performance at different developmental stages was evaluated. The results indicated that both species possess a similar gonadosomatic index. The parent *C. gariepinus* possesses significantly (*p* < 0.05) higher male reproductive quality than *C. macromystax*. Fecundity was significantly higher (*p* < 0.05) in the female parent of *C. macromystax* and lower in *C. gariepinus*. The highest fertilization rates and hatching rates were achieved in the hybrid cross of ♀*C. macromystax* × ♂*C. gariepinus* (♀*Cm* × ♂*Cg*) which also recorded the lowest deformed larva rate. Survival rates at the larval stage were more than 80%, with the highest record of survival in the hybrid ♀*Cm* × ♂*Cg*. However, survival rates in the fry stage were lower than 70%. The hybrid cross (*C. macromystax* × *C. gariepinus*) outperformed the parental cross of *C. macromystax* but was not significantly similar to the parental cross of *C. gariepinus*. The study revealed the potential of the hybrid ♀*C. macromystax* × ♂*C. gariepinus*, and hybridizations of ♀*C. macromystax* × ♂*C. gariepinus* (♀*Cm* × ♂*Cg*) are indeed possible and proved to have a better growth and survival rate under semi-arid conditions, which would contribute significantly to the improvement of *C. macromystax* production in captivity.

## 1. Introduction

*Clarias gariepinus* (Burchell, 1822) and *C. macromystax* (Gunther, 1864) are two of the most popular *Clariidae* species in Nigeria, and as a result, they have a high commercial value [1,2]. The minimum number of African catfish required in Nigeria is 4.3 billion, with a total supply of 55.8 million available from all sources (FDF, 2007). As a result, Nigerian catfish fingerling production is insufficient to meet the demand from fast growers [3,4]. *C. gariepinus* is more widely available, whereas *C. macromystax* is rare and commands a high price due to its good taste, attractive appearance, and eel-like motion. *C. macromystax* is in high demand, but its fingerlings are limited due to challenges in the production of seeds in captivity [5,6]. Both *Clarias* species have significant potential to assist Nigeria’s rapidly growing aquaculture industry, but due to a scarcity of fish seeds (fingerlings), Nigeria’s goal of reducing animal protein deficiency with fish would be a mirage [7]. Fish farmers and consumers have long wished for a farmed catfish that combines the two characteristics of *C. macromystax* spp. with the fast growth rate of *C. gariepinus* [8,9]. There is a paucity of reports on *C. macromystax* potentials of culture in captivity and an estimate of broodstock yield through gonadosomatic indices, partly due to the high cost of the fish [10,11,12]. The current study was designed to evaluate the reproductive traits of *C. gairepinus* and *C. macromystax* and their relationship with broodstock sizes [13,14].

Many countries around the world rely on fish as a protein source, a livelihood, and a source of foreign exchange. Global (world) annual increases in fish consumption are (3%) more than the (2%) yearly increase in population, rising from 10.2 to 20.50 kg per capita [11,15,16]. Artisanal fisheries have remained stable over the last ten years, and the development of adequate skills, knowledge, and technology to domesticate, maintain, feed, and breed various fish species across the various aquaculture businesses has been critical to the rise of global aquaculture [2,4,10].

The introduction of *C. gariepinus* for intensive culture in concrete tanks and earthen ponds by some Dutch consultants, along with the influx of imported extruded catfish feeds, represented a watershed moment in the Nigerian aquaculture industry’s history [11]. Farmers were enticed to abandon the native strains that were already being domesticated because of their rapid development and large harvests [12,13].

Fecundity, or reproductive capacity, is a term that refers to a critical biological metric used to determine a fish stock’s commercial viability [14,15]. The deteriorating performance of domesticated *Clarias* species with the decline in growth and reproduction as well as appearance of deformities is raising concern among fish farmers [2,17]. *Clarias gariepinus* and *C. macromystax* (*Claridae*) are extremely popular among fish farmers for their highest pricing on the market in Nigeria, rapid growth rate, and ability to survive adverse pond conditions such as low oxygen and high turbidity [18,19,20]. *C. gariepinus* fry culture as a source of fish seed is increasingly becoming essential since the fish contributes to family health, income generation, and work prospects due to food abundance and nutritional advantage [17,19].

Evaluations of reproductive methods are key components of the fish species that are being studied to learn more about their biology and population dynamics [10,21]. Fecundity may vary as a result of environmental adaptations [9,14]. Combining this information with estimations of the generation of eggs enables the estimation of the biomass of spawning stocks [15,17]. Even among members of the same species, fecundity might vary because of behavioral and spawning differences [22,23]. Selecting the proper strains is crucial for a successful breeding program not only to meet the production target but also to cut costs, and increase disease resistance, feed resource use, and product quality [20,24].

The hybridization research has been prompted by the need for high-quality fish seed; currently, there is no research on the fecundity and reproductive traits of *C. macromystax* and *C. gariepinus,* which are the two most important species in Nigeria [15,21]. To evaluate the reproductive parameters such as fertility rate, milt volume, the weight of the ovary, fertilization, and hatchability of *C. macromystax* and *Clarias gariepinus* in Nigeria [18,20].

The objective of the study was to evaluate the reproductive parameters of *C. gariepinus* and *C. macromystax* under the semi-arid conditions of the northeastern part of Nigeria.

## 2. Materials and Methods

### 2.1. Experimental Site

The study was performed in the Department of Fisheries hatchery complex at the University of Maiduguri, which is located between latitude 110°48′51.45″ N and longitude 13°11′48.116″ E at an elevation of 375 m above sea level (Google Maps 2018). The study region receives an average of 600 mm of rainfall per year, with the hot season lasting from March to July and Harmattan lasting from November to February. The rainy season begins in June and finishes in October.

### 2.2. Collection Area

*C. gariepinus* gravid males and females were collected from Gubi Dam in Bauchi State, which is located between latitude 10′42′ N and longitude 9′87′ E, and Lower River Benue, Adamawa State, which is located between latitude 9′28 N and longitude 12′26 E. The readiness of the genitals was used to select the brood stocks; gravid females were chosen based on swollen, reddish genital openings, while gravid males were chosen based on reddish and pointed genital papillae. The broodstock was transported to the Department of Fisheries hatchery unit in 25 L Jerrycans cut in half horizontally. For two weeks, the broodstock were acclimatized and conditioned in a 7 × 5 × 1.2 m polyethylene-lined earthen pond and fed twice daily.

### 2.3. Brood Stock Selection Inducement

Six females from each species were selected and transported to the hatchery. They had a large and soft belly, a projecting reddish genital papilla, and oocytes of a greenish color that were easily collected (after being induced with Ovaprim hormone) by light physical pressure on the abdomen [15,16]. Oocytes were collected from each female in the hatchery using a plastic cannula, and their diameters were measured on millimetric paper. Six females from each species were selected and housed singly in 20 L plastic containers with an oocyte diameter of between 1.1 and 1.6 mm for *C. gariepinus* [16] and between 1.65 and 1.73 mm for *C. macromystax*. Each of these containers was assigned a unique number based on the species, and each container was connected to a 2 L/min water replenishment system. Three males of each species weighing more than 200 g were chosen for their well-developed genital papillae [16]. All males from each species were housed in identical concrete tanks. In one of the tanks, a mercury thermometer was inserted for temperature management. Figure 1 illustrates some of the exterior characteristics of the broodstock that was chosen.

### 2.4. Determination of Fecundity

Fecundity (the number of eggs produced per kilogram of female body weight) is determined by the number of quality eggs produced by each species after the latency period. The female fish was induced (with the hormone Ovaprim) and weighed using an electric weight balance to the nearest gram, stripped into a dry plastic bowl, and the eggs were weighed to calculate the number of eggs from each egg mass of the female. A one gram sample was taken from each egg mass and fixed in buffered (10%) formalin for 12 h then transferred to 70% ethanol for storage before counting in a calibrated Petri dish using a tally counter under a dissecting microscope at ×20 magnifications volume. The number of eggs spawned was calculated by multiplying the weight of the egg mass (from each female) by the number of eggs present in 1 g of the respective egg mass.

### 2.5. Total Egg Number Release

To calculate the total egg number released by each female, the weight of the broodstock after it has been stripped was subtracted from the total weight of the broodstock before stripping (Wa), and the difference was multiplied by the number of egg counts per gram. The formula is as follows:Number of eggs released = (Wb − Wa)g × N
where Wa is the weight of broodstock before stripping; Wb is the weight of broodstock after stripping; g is gram; and N is the number of eggs/1 g.

### 2.6. Determination of Gonadosomatic Index

Gonads were collected and weighed from females and males of each species. The testes and egg lobes were removed and weighed using a delicate balance. Dissection of male broodstocks was performed to remove the testes. After dissection, both the left and right testes were retrieved, weighed, and the semen collected. Each testis’ gonadal weight and volume of sperm were determined. After obtaining the testicular weights, a longitudinal incision was performed in each testis, and the milt was collected in calibrated glass tubes to determine the semen volume. After sample activation with a hypertonic solution, the motility of the sperm cell in the milt collected was studied under the microscope. Milt was pressed and rinsed in physiological saline that had been prepared previously. Milt was then sieved to eliminate dead tissues. The gonadosomatic index (GSI) was calculated based on the following formula:$$\mathrm{Gonadosomatic}\;\mathrm{Index}\;(\mathrm{GSI})=\frac{\mathrm{Weight}\;\mathrm{of}\;\mathrm{Gonad}\;\left(g\right)}{\mathrm{Eviscerated}\;\mathrm{Weight}\;\left(g\right)}\;\times \;100$$
The condition factor K = (W/L3) × 102
where W = weight (g) and L = length (mm).

### 2.7. Estimation of Fertilization and Hatching Rate

Fertilization and the hatchability rate in this study were determined using 1 g (183–200 oocytes) of eggs from each cross; the egg number was estimated using the gravimetric method (eggs per gram); and 1 g of eggs from each species was used to determine the fertilization rate. The 12 bowls are labeled corresponding to each crossing combination, including the control plastic bowl. Each had a UV light and a pump. The height of the water level is 0.35 m, and the flow of water is 5.18 mL^−1^. This was used to incubate the eggs, which were kept at an average temperature of 27 °C in the hatchery. The time spent waiting for the eggs in the control bowl to turn white was recorded. The eggs containing embryonic eyes 45–60 min after fertilization were considered fertilized and counted to estimate the fertilization rate. The number of hatchlings per container was recorded by direct counting of the hatchlings and unhatched eggs for each cross combination. At the end of hatching, deformed or dead larvae were counted and siphoned off to distinguish them from normal larvae. The larvae were counted directly with the naked eye during the day. A thermometer was used to record the water temperature in the rearing tanks every day in the morning, noon, and evening. The following formulas were used to calculate the rates of fertilization, hatching, and larval survival:(i)Fertilization rate (%) = Number of fertilized eggs/Number of estimated eggs × 100;(ii)Hatchability (%) = $\frac{\mathrm{Total}\;\mathrm{Number}\;\mathrm{of}\;\mathrm{hatched}\;\mathrm{eggs}}{\mathrm{Total}\;\mathrm{Number}\;\mathrm{of}\;\mathrm{fertilized}\;\mathrm{eggs}}$ × 100;(iii)Survival (%) = $\frac{\mathrm{Total}\;\mathrm{Number}\;\mathrm{of}\;\mathrm{larvae}\;-\;\mathrm{Number}\;\mathrm{of}\;\mathrm{dead}\;\mathrm{larvae}}{\mathrm{Total}\;\mathrm{Number}\;\mathrm{of}\;\mathrm{larvae}}$ × 100%.

### 2.8. Determination of Water Quality

Temperature, dissolved oxygen concentrations, total hardness, total alkalinity, and the pH of the water in the experimental troughs were determined using probes and the LaMotte Fresh Water analysis test kit, model AQ2/3.

### 2.9. Statistical Analysis

A one-way analysis of variance (ANOVA) was used to analyze the statistical data gathered from the experiment. A Duncan Multiple Range Test (DMRT) was employed to determine the differences in means (*p* = 0.05). The descriptive statistics mean, standard deviation, and percentage were used to describe the data.

## 3. Results

The results in Table 1 summarize the fecundity of the two *Clarias* species, and fecundity varied significantly (*p* < 0.05) between the two *Clarias* species, with *C. gariepinus* having the highest fecundity and *C. macromystax* having the lowest fecundity. *C. macromystax* produced substantially more oocytes per gram of egg mass than *C. gariepinus*. All parameters tested were statistically different (*p* < 0.05) between the two species.

Table 2 indicates the gonadosomatic indices for *C. gariepinus* and *C. macromystax*. The results of the study revealed that the female *C. gariepinus* fecundity rate was very high compared to *C. macromystax* broodstock. The weight of the ovary differed significantly (*p* < 0.05) between *C. gariepinus* and *C. macromystax*.

The mean volumes of sperm and testes motility from *C. macromystax* and *C. gariepinus* are listed in Table 3. The *C. gariepinus* males had the highest volume of milt (9.5 mL) compared to the *C. macromystax* (7.5 mL), testes weight (8.90 and 6.70 g), motility (63.0 and 55.0 s), and testes length, i.e., both left and right (4.74 and 3.80 cm; 6.55 and 4.55 cm). The outcome of the results demonstrates that there is a significant (*p* < 0.05) difference in males from the cross between *C. gariepinus* and *C. macromystax*.

The male reproductive features of adult *C. gariepinus* and *C. macromystax* broodstocks are similar except the weight of the right testis. The trend indicates that *C. gariepinus* has a considerably (*p* < 0.05) greater male reproductive quality than *C. macromystax*, as indicated (Table 4) by a higher gonadosomatic index in *C. gariepinus*. As indicated in Table 5, the reproductive characteristics of female broodstock of *C. gariepinus* and *C. macromystax* were statistically (*p* > 0.05) similar. Despite the apparent higher live weight of *C. gariepinus*, which may result in a higher apparent egg weight and egg population per female, *C. macromystax* had a higher gonadosomatic index.

The results of the fertilization and hatching rates of *C. gariepinus, C. macromystax,* and their hybrids are presented in Figure 1.

The result of the fertilization and hatching rates varied significantly amongst cross combinations, as illustrated in Figure 1. The highest fertilization rate (79.23%) was seen in the pure progeny of *C. macromystax* (*Cm* × *Cm*), followed by a (65.93%) fertilization rate in the reciprocal hybrid cross of *Cm* × *Cg*, while the hybrid *Cg* × *Cm* had a fertilization rate of 61.00%, and the pure progeny cross of *Cg* × *Cg* had a fertilization rate of 50.00%. There were substantially significant differences in fertilization levels between the pure progeny of *C. macromystax* (*Cm* × *Cm*) and the reciprocal hybrid of *C. macromystax* × *C. gariepinus* (*Cm* × *Cg*) when compared to the *C. gariepinus* × *C. macromystax* (*Cg* × *Cm*) hybrid and the pure progenies of *C. gariepinus* (*Cg* × *Cg*).

The hatching rate of reciprocal hybrids *C. macromystax* × *C. gariepinus* (*Cm* × *Cg*) was (*p* > 0.05) significantly higher (72.40%) in all crossing combinations, while the lowest hatching rate was obtained in the pure progeny of *C. macromystax* × *C. macromystax* (*Cm* × *Cm*). Female *C. gariepinus* × male *C. gariepinus* had the highest rate of hatchability (88.0%), while the lowest hatching rate was obtained in the cross between female *C. macromystax* × male *C. gariepinus* (77.0%). However, there were no significant differences in the hatching rates between the pure progeny of *C. gariepinus* (*Cg* × *Cg*) and the *C. gariepinus* × *C. macromystax* (*Cg* × *Cm*) hybrid. The rate of larvae that are deformed in C. *macromystax* and *C. gariepinus is* illustrated in Figure 2, which reveals that it varied from one cross combination to another in all the crossing combinations. The deformed larvae rate was significantly (*p* < 0.05) lower in the reciprocal hybrid of *C. macromystax* × *C. gariepinus* (*Cm* × *Cg*). There were no significant differences in the deformed larvae between the hybrid cross of *C. gariepinus* × *C. macromystax* (*Cg* × *Cm*) and the two pure progeny of *C. gariepinus* and *C. macromystax*.

## 4. Discussion

The higher hatching and fertilization rates obtained in this study contradict the findings of [8,10], who reported similar results after successfully crossing exotic and native *C. gariepinus*. According to [21], reciprocal crosses between *H. longifilis* and hatchery strain C. *gariepinus* resulted in lower fertilization rates (86–89%); however, the pure [25] had the greatest fertilization rate (95–100%), which differed from [25,26], who reported a higher mean fertilization rate of 75.75% for the *H. birdosalis* cross with *C. gariepinus*, 75.49 percent for the pure Jamuna strain, and 65.49 percent for the pure hatchery strain. It is possible that the reduced fertilization rate in the crosses is related to differences in populations. Additionally, ref. [27] hybrid crossings between *C. gariepinus* and *C. batrachus* obtained a low fertilization rate of (77.10%) between the four intra-specific crossbreeds. However, the pure parental crosses were shown to have a higher hatchability rate in the study. This hatchability result between the pure *C. gariepinus* and *C. macromystax* and their reciprocal hybrids is consistent with the finding of [28], who reported a lower hatchability rate (41.0%) among the reciprocal hybrids as compared to the (94.0%) pure crosses of the parents. The authors of [29] reported similar results when crossing female exotic × male local *C. angularis* (52.1%) and local × exotic *C. gariepinus* (49.5%). It is crucial to note, however, that variances in breeding histories can be influenced by water quality and the age of the fish (particularly hatching and fertilization rates, as reported by [30,31,32]. Seasonal changes can also lead to disparities in hatching and fertilization rates, as reported by [20,31].

The results of this study revealed that females of *C. gariepinus* had the highest fecundity rate (198,205 eggs), which coincides with [26,30], who both reported that most larger fish have a greater fecundity rate compared to smaller fish. According to him, the size of eggs increases as the number of eggs in a species decreases; thus, similar results were discovered by [13,15] that fecundity was also affected by fish size; the larger the fish, the more eggs it produced, possibly due to more visceral volume available to hold the eggs. The right (4.74 cm) and left (6.55 cm) testes length with the volume of the milt (9.5 mL) are significantly (*p* < 0.05) higher in *C. gariepinus* compared to the broodstock of *C. macromystax*, which has shorter testes length (right 3.80 cm and left 4.55 cm) and a smaller milt volume (7.5 mL), respectively. The differences could be attributed to the weight of the fish. *C. gariepinus* had a weight of 1500 g, compared to *C. macromystax* with a weight of 550 g.

The stocking density, sex ratio, size, age, and feeding quality may all influence egg and sperm quality, as reported in [14]. Several factors, including feeding regime, feed quality, and milt volume and count, have been demonstrated to influence qualitative aspects of the milt (milt volume, sperm lobe length, sperm motility, and count) [19]. Environmental influences, individual differences age, weight, and fish length season of the year [30], stress, intake of nutritive and genetic resources, and physiochemical parameters of water (pH, salinity, temperature, and dissolved oxygen) were all mentioned by [28,31]. Temperature, dissolved oxygen, and pH during the breeding phase, on the other hand, coincide with [32,33].

The trend of the results indicates that *C. gariepinus* has a higher gonadosomatic index than *C. macromystax*. Superior testicular diameters in *C. gariepinus* increased the amount of sperm collected from males. The volume of semen recorded in *C. gariepinus* (16.69 vs. 1.49 mL) indicates that fewer males will be required to fertilize eggs in cultured breeding in comparison to *C. macromystax*, which requires more males to fertilize eggs. The authors of [30] observed that testis size is an excellent indicator of spermatogenesis efficiency, implying that *C. gariepinus* has better spermatozoa production efficiency. Despite the apparent higher live weight of *C. gariepinus*, which results in a higher apparent egg weight and egg population per female, *C. gariepinus* had a higher gonadosomatic index. Due to the identical female reproductive characteristics of both species, difficulties in producing *C. macromystax* catfish seeds in captivity are likely due to male gonadal deficits. The authors of [26] also found that male *H. longifilis* had a lower gonadosomatic index than females (0.91–0.87 in males vs. 2.34–2.22 in females). This indicates that breeding *C. gariepinus* is less expensive than breeding *C. macromystax*, looking at the high market value of *C. macromystax* due to its good taste when the cost of males is included. Given the economic importance of *C. macromystax* in the catfish market, it is critical that fish breeding objectives focus on boosting the species’ reproductive performance. The scarcity of *Clarias macromystax* fingerlings is explained by the male’s low gonadosomatic index, as discovered in this study.

According to the findings of this study, in male *C. gariepinus*, total gonadal weight, whole right and left testis weight, right testis semen volume, and total gonadal weight are all significantly correlated with gonadosomatic index. This is in contrast to the findings of [8], who found that the live weight of a male had a strong positive correlation with the weight of his testicles in *C. gariepinus*. According to the findings of this study, female *C. gariepinus* body measurements can be used to predict their live weight. This trend is consistent with the findings of [8,16], who discovered that larger broodstock catfish had better breeding traits in *C. gariepinus*. According to the findings of this study, efforts to influence the increased gonadosomatic index of male *C. macromystax* could be directed toward enhancing gonadal development. The trend indicates that the live weight of female catfish (*C. gariepinus* and *C. macromystax*) can be predicted by their body measurements, particularly their standard length and total length. According to the findings of this study, female *C. macromystax* can be identified by their body measurements, which can be used to predict their live weight. In contrast to [24], which found a positive correlation between female egg weight, fertility, and hatchability, this study found a negative correlation between female egg weight, fertility, and hatchability. In the case of male catfish (*C. gariepinus* and *C. macromystax*), the reproductive indices can only be used to predict and influence the gonadosomatic index, not the other way around. Possibly because of the claims made by [33] that the size of the testis is a good predictor of spermatogenesis efficiency.

## 5. Conclusions

According to the results of this research, male reproductive features in the two catfish broodstocks differed depending on the species, while female reproductive capability in *C*. *gariepinus* and *C. macromystax* was similar regardless of their species. The gonadosomatic index and reproductive characteristics of male *C. gariepinus* were shown to be significantly superior to those of *C. macromystax*. The poor reproductive performance of male *C. macromystax* broodstocks may be a contributing factor to the cause of the species’ fingerling scarcity and difficulties in seed production, which have been attributed to *C. macromystax* species.

## Figures and Tables

**Figure 1 animals-13-01723-f001:**
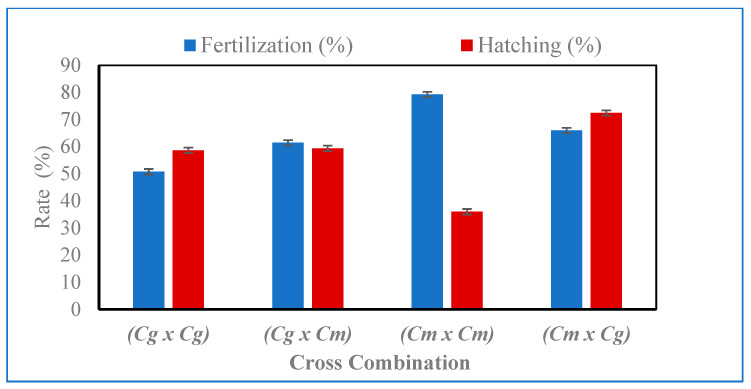
Fertilization and hatching rates in pure progenies of *C. gariepinus*, *C. macromystax,* and their reciprocal hybrids under semi-arid conditions.

**Figure 2 animals-13-01723-f002:**
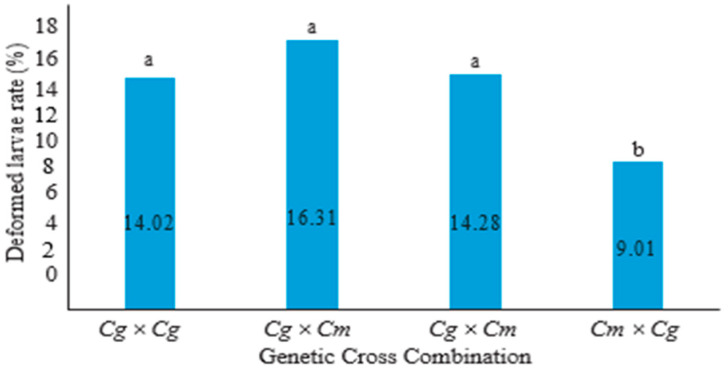
Deformed larvae in progenies and reciprocal hybrids of *C. gariepinus* and *C. macromystax* under semi-arid conditions; a and b: histograms with identical letters are not significantly different (*p* > 0.05).

**Table 1 animals-13-01723-t001:** Fecundity of *Clarias macromystax* and *Clarias gariepinus* in this study.

Parameter	*C. gariepinus*	*C. macromystax*
Female weight (kg)	1.13 ^a^	0.85 ^b^
Number of eggs (1/g^−1^)	700 ^a^	845 ^b^
Fecundity (eggs/kg^−1^ fish)	145,715.17 ± 1283.51 ^b^	93,672.50 ± 477.92 ^a^

The mean values for identical letters a, b in a row are not statistically significantly different (*p* > 0.05).

**Table 2 animals-13-01723-t002:** Mean fecundity and gonadosomatic index of female *C. gariepinus* and *C. macromystax* under semi-arid conditions.

Strains	Weight (g)	Total Length (cm)	Weight of Ovary (g)	Number of Eggs	GSI (%)	PGSI
*C. gariepinus*	1310.0 ^a^	51.0 ^a^	192.8 ^a^	198,105.0 ^a^	12.92 ^b^	17.41 ^b^
*C. macromystax*	651.0 ^b^	43.0 ^a^	132.1 ^b^	163,054.0 ^b^	16.89 ^a^	25.51 ^a^
Standard Error of the Mean	0.57 *	0.57 ^ns^	0.58 *	0.57 *	0.06 *	0.06 *

Means with the same superscript a and b in each column are not significantly different (ns) (*p* > 0.05). * stand for significant difference.

**Table 3 animals-13-01723-t003:** Mean semen motility, sperm weight, and volume from *C. gariepinus* and *C. macromystax*.

Strains	Weight (g)	Length (cm)	Testes Weight (g)	Length of Right Testes (cm)	Length of Left Testes (cm)	Milt Volume (mL)	Motility Duration (s)
*C. gariepinus*	1500.0 ^a^	56.40 ^a^	8.90 ^a^	4.74 ^a^	6.55 ^a^	9.50 ^a^	63.00 ^a^
*C. macromystax*	550.0 ^b^	49.75 ^b^	6.70 ^b^	3.80 ^b^	4.55 ^b^	7.50 ^a^	55.00 ^b^
Standard Error of the Mean	28.86 *	0.05 *	0.58 *	0.01 *	0.58 *	0.57 ^ns^	0.57 *

Means with the same superscript a, b, and ns in each column are not significantly different (*p* > 0.05). * stand for significant difference.

**Table 4 animals-13-01723-t004:** Gonadosomatic index of *C. gariepinus* and *C. macromystax* with their testicular characteristics.

Parameter	*C. macromystax*	*C. gariepinus*	*p*-Value
Weight of the right whole testis (g)	13.21 ± 4.61	2.64 ± 1.33 *	0.04
Weight right testis	4.81 ± 1.73	1.52 ± 0.53	0.08
Volume right testis semen (mL)	8.41 ± 3.08	1.12 ± 0.80 *	0.03
Weight of the left whole testis (g)	13.40 ± 2.55	1.45 ± 0.32 *	0.00
Weight of the left testis (g)	4.82 ± 1.30	1.09 ± 0.17 *	0.01
Volume left testis semen (mL)	8.58 ± 1.92	0.37 ± 0.18 *	0.00
Total semen volume (mL)	16.69 ± 5.58	2.49 ± 0.98 *	0.01
Total testis weight (g)	10.25 ± 3.36	2.60 ± 0.67 *	0.03
Liveweight (kg)	2.54 ± 0.12	2.11 ± 0.15 *	0.04
Total gonadal weight (g)	27.94 ± 8.17	4.09 ± 1.63 *	0.01
Gonadosomatic Index	1.15 ± 0.38	0.19 ± 0.06 *	0.02
Condition factor	0.84 ± 0.07	0.66 ± 0.06	0.01

* Data are presented as n, the number of samples, and a mean ± standard deviation.

**Table 5 animals-13-01723-t005:** Gonadosomatic index and egg output of *C. macromystax* and *C. gariepinus*.

Parameter	*C. macromystax* (n = 20)	*C. gariepinus* (n = 20)	*p*-Value
Live weight (kg)	2.07 ± 0.11	3.66 ± 1.18	0.08
Weight of eggs (g)	166.67 ± 31.69	233.33 ± 55.48	0.30
Egg population (×10^3^ eggs/female)	77.28 ± 22.52	94.08 ± 44.63	0.72
Gonadosomatic Index	8.24 ± 1.62	7.04 ± 1.64	0.66
Condition factor	0.85 ± 0.13	0.85 ± 0.08	0.32

## Data Availability

The study did not report any data.

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
