# Peer review of "Evaluation of Fecundity, Fertilization, Hatching, and Gonadosomatic Index of Exotic Clarias gariepinus (Burchell, 1822) and Native Clarias macromystax (Gunther, 1864) under Semi-Arid Conditions of Nigeria"

_animals, 2023, doi:10.3390/ani13111723_

Round 1

Reviewer 1 Report

A very interesting article in terms of content. However, it needs a very thorough improvement. I included my detailed comments in the text. To view them, open the attachment in Acrobat Reader. In my opinion, the article should be resubmitted after taking into account the corrections. MS has potential, but it's hard to assess it, because some figures are missing, tables and figures are not numbered correctly, sentences are ended with commas, etc .... I also think that the experimental scheme helped the readers to understand this work.

Reviewer 2 Report

This is a well-performed and interesting study, providing new data on reproductive parameters and hybridizations between two African catfishes (C. gariepinus and C. macromystax). Results show the viability and good performance of gamete crosses between the two species, through the analysis of fertilization and hatching rates, larvae malformations and survival and growth rates until fry stage. Data has both scientific and technological interest. The study is well written and presented. I have only some minor comments.

Minor comments

Methods (l. 123): what do you mean with ”….after the latency period from each species.”.

Methods (l. 123): You say “The induced female………….”, what does it means?, there is no previous mention to any induction protocol.

Methods, section 2.7: provide more details on the in vitro fertilization protocol, such as ratio sperm/egg, incubation time, etc.

Table 3.1. Indicate in the legend how data are expressed (mean+-SEM?) and the n.

Round 2

Reviewer 1 Report

MS has been significantly improved, but still requires minor revision. My comments/suggestions are included in the text. To see them all, open the file in Acrobat Reader. I think that Aurors will have no problems with improving this MS.

Author Response

We are very grateful for your contribution towards improving the quality of our manuscript, The response to the comment was highlighted with yellow color below are responses to your comment.

S/No

Reviewer 1 Comment

Response to Reviewer 1 Comment

1.

You  probably used artificial reproduction. If so, there is no information on the temperature of the water and the hormonal agent used (and possibly information on the anesthetic`, if any were used by the authors). Please use publication citations when adding this information, e.g. Animal Reproduction Science, 2021, 231, 106798 for artificial reproduction; e.g. Animal Reproduction Science, 2019, 211, 106222 for fertilization and e.g. Diseases of Aquatic Organisms, 2018, 127(3), pp. 237–242 for the assessment of embryo deformities.

The Hormone agent was added in line 135, and anesthetic was not used in the study.

2.

This looks like an unfinished sentence - please correct.

All sentence is corrected in line 235, 236.

It should be corrected using authors initials

The authors initials was included in line 354, 355,356,357,358

The list of references should be in the order of citations in the text, not alphabetically.

The reference list is corrected in the order of citation.